## [Peer Review File · Nature Communications]

Reviewers' comments:

Reviewer #1 (Remarks to the Author):

The article entitled "Buoyant particulate strategy for few-to-single particle-based plasmonic enhanced nanosensors" is the result of an improvement in the nanoparticle distribution after the drying process of a suspension hoping that this final distribution is much better organized than the so-called coffee ring process. Although the results are interesting and encouraging for the development of highly sensitive optical sensors, there are some dark issues that strongly recommend the publication in a more specific journal. Among these issues I can mention the followings:

1. It is mentioned that "The most superiority of current strategy is the aggregation effect of the buoyant-particulate on slippery Surface. As a result, the solvent and containing analyte can also be enriched into the vicinity of buoyant-particulate aggregate". This implies that the presence of the large silica particles induces a redistribution of the nanoparticles, but it is doubtful that this improved method could further enhance the Raman or fluorescence intensity by a mere plasmonic effect.
2. The experimental part is not clear about the way through which the laser beam is placed in the gap between particles, and this should be clarified in the manuscript.
3. The section entitled "Droplet drying processes in buoyant particulate system" is too long and very much unclear, and also a long explanation is devoted in the supplementary materials.
4. The probe molecules employed in the so-called "Plasmonic sensing properties" were mainly colorants, such as CV or R6G. These molecules give rise to very high sensitivity (10-18 M). Additionally, other more interesting molecules such as POPs rendered also a high sensitivity (less than ppb). This sensitivity clearly is in the range of the single molecule detection. It is difficult to believe that such a high sensitivity could be got only by a simple rearrangement of the nanoparticles from coffee ring to large particle gaps where a large plasmonic effect is not expected at all because of the macroscopic character of this nanoparticle redistribution.
5. Finally, another discussible issue of this paper is the fact that no bands coming from the chemical reactive employed in the fabrication of the nanoparticles and the linking system did not appear. For instance, the citrate bands, the AMPTS, etc. did not appear while others like Bisphenol A appears at a very strong intensity.

Reviewer #2 (Remarks to the Author):

In this manuscript, the authors report a strategy to enrich and trap an analyte at a "hot-spot" for increased SERS and fluorescence sensitivity. For this they utilize, dynamics in a drying droplet. Although observed phenomenon and its utility for the goal is interesting, the conclusions are stretched too far. First of all, such sensitivity can be achieved on silver nanoparticles (AgNPs). The second issue is that the test analyte, crystal violet (CV). It is a fact that this molecule has tremendous SERS activity and the results with this molecule without testing another analytes, is quite misleading. As the authors stated themselves, this is one of the other proof of concept reports that may not go too far.

Therefore, I cannot recommend the acceptance of this manuscript in this journal.

Reviewer #3 (Remarks to the Author):

The work by Zhang et al reports a plasmonic sensor, which harnesses the electromagnetic hotspots generated through buoyant plasmonic particulate strategy. This particle assembly strategy avoids the coffee-ring effect and is able to guide the target molecules into the hotspot regions to take full advantage of the enhanced electromagnetic field to achieve highly sensitive detection of various molecules. However, the method is not practical for real-world application owing to the difficulty in accurately controlling the volume and to place the particles inside the solution. The work described in the manuscript does not significantly advance the state-of-the-art. It might be better suited for a more specialized journal (e.g., *Light: Science and Applications*) There are also several suggestions listed as follows:

1. The particulate system is not well characterized. What is the structure of the particles before and after Au deposition? SEM images and TEM images are required to show the buoyant plasmonic nanostructures and to show the size of the AuNPs on the buoyant particle. What are the optical properties of such particles? For example, localized surface plasmon resonance wavelength of the particles critically determines their SERS activity for a given excitation wavelength. So the authors should better characterize the materials. How does the size of the AuNP effect the final enhancement efficiency?
2. The conclusion in line 97-100 in page 5 needs more experimental support. It is not convincing to state that CV molecules do not absorb on the substrate but in the vicinity of particle-particle interface by simply showing the absence of fluorescence signal after removing the particles. In fact, this argument is one of the main advantages the author claimed in the paper. However, the non-detectable fluorescence signal on the substrate might due to the absence of plasmonic enhancement.
3. Fluorescence quenching effect needs to be considered when calculating the fluorescence enhancement factor (page 5, line 111-114) by comparing the signal between the interface and non-interface regions. Due to the direct contact between the fluorophores and the Au surface, fluorescence signal from the fluorophore might be quenched. Therefore, the measured enhancement factor might need to be modified.
4. Other factors that affect the drying process might be interesting to know for the readers.
5. Error bars are missing in several important plots, such as Figure 4d. The experiment needs to be repeated at least three times to show the reproducibility and determine the coefficient of variance. In other words, what is the sample-to-sample variation in this sensing system?
6. Scale bars are missing in Figure 2 (a) (b), Figure 3 (b), Figure 4 (b)
7. Overall, the manuscript is not well-written and makes the readability somewhat difficult. The authors should carefully proofread and minimize grammatical errors.

Reviewers' comments:

Reviewer #1 (Remarks to the Author):

The article entitled “Buoyant particulate strategy for few-to-single particle-based plasmonic enhanced nanosensors” is the result of an improvement in the nanoparticle distribution after the drying process of a suspension hoping that this final distribution is much better organized than the so-called coffee ring process. Although the results are interesting and encouraging for the development of highly sensitive optical sensors, there are some dark issues that strongly recommend the publication in a more specific journal. Among these issues I can mention the followings:

1. It is mentioned that “The most superiority of current strategy is the aggregation effect of the buoyant-particulate on slippery Surface. As a result, the solvent and containing analyte can also be enriched into the vicinity of buoyant-particulate aggregate”. This implies that the presence of the large silica particles induces a redistribution of the nanoparticles, but it is doubtful that this improved method could further enhance the Raman or fluorescence intensity by a mere plasmonic effect.

Response: we appreciate reviewer-1 for the acknowledgement of the work with the input of “results are interesting and encouraging for the development of highly sensitive optical sensors”. Owing to that manuscript is not well-written and makes the readability somewhat difficult, therefore, it seems we make reviewer-1 confused. In fact, the main idea is not “large silica particles induces a redistribution of the nanoparticles”, but the enrichment and localization effect of the detection molecules. Actually, in the supporting information in the initial version, we have showed the structure of the buoyant particulate as follows. We can see that the Au nanoparticles initially coated on the silica surface by the Au seed decoration and growth processes. Certainly, in the revised version, we redited the Extended Data Fig. 1, 2, and 3, to present clearly.

Extended Data Fig. 1 | The schematic for the preparation process of buoyant SiO₂-Au core/shell particulate, via Au seeds decoration and growth of Au nanoparticles on silica core.

With this buoyant particulate strategy, we may obtain below several significances,

(1) Our study represents a significant new concept on how to avoid the coffee ring phenomenon for the application of nanoparticles.

Coffee-ring effect is a very common phenomenon. In 1997 (nature, 1997, 389, 828), researcher reported the nature of coffee ring, which is the capillary flow outward from the centre of the

drop carries dispersed particulates to the edge as evaporation proceeds. In 2011 (nature, 2011, 476, 308), researcher found the “suppression of the coffee-ring effect by shape-dependent capillary interactions”.

Why we are talking about the coffee-ring effect here?

Since for the applications of NPs, such as in any kinds of nanosensing used for molecule detection, the uniformity of a substrate is so so important. See above example (RSC Adv, 2013, 3, 17829). From A, for 1, 2, and 3 points, point-3 shows better signal, while, for B, normally point-1 should be better than point-2, but the results are quite different. The signal repeatability is a main problem for many kinds of applications of NPs, which is one of the key factor that block their practical applications.

Therefore, in the last decades, you may find **so many researches were carried out to fabricate array structures** by means of nanofabrication routes such as lithography or template. On the other hand, the **self-assembly of NPs** is also to solve the structural uniformity issue when using NPs as building units.

In plasmonic sensing researches, including SERS detection, the formation of coffee-ring may result in completely uncontrolled distribution of both colloidal nanoparticles and target molecules, thus a worse signal uniformity and sensitivity.

In this work, large-sized, light-weight floatable particles are designed. During the drying process, the buoyant particles may float on the top of the solvent, thus significantly reduce the chance of being pinned on slippery surface. The buoyant particulates, like a micro-sized “boat”, finally land on the substrate. **As a result, the coffee-ring phenomenon has been fully avoided.**

(2). The enrichment or localization of detected/target molecules is very important topics for many important sensing applications.

In current work, we used large-sized floating particles combining with slippery surface may enhance the spatial enrichment and localization capability of the analyte into plasmonic sensitive sites via the aggregation and lifting effect, as shown in below fluorescent images.

As a result, we achieved an excellent enrichment capability with an **enrichment factor up to an order of $\sim 10^4$** . That is to say, we concentrate the molecules initially in a 0.5 ml solution and localize to the few tens micrometer area as shown in below images.

(3). The property of current sensing strategy is very sensitive, i.e., for enhanced Raman and fluorescence spectroscopies.

Due to the excellent enrichment capability, the limit of detection for most of molecules may down to **fM level with 100% probability** to collect observable signal. Even for the **typically weakly adsorbed molecules, a limit of detection of ~ 0.1 ppb level has been easily obtained**. These results are quite few if compared with other techniques in the literatures, even at least three or four orders of magnitude higher than that of marketing product.

Actually, the signal enhancement capacity is composed of the contributions from enrichment factor of $\sim 10^3$ - 10^4 and the plasmonic SERS enhancement factor of $\sim 10^7$ - 10^8 .

(4). We used this significant concept and methodology of this work with a very facile, simple, and cost-saving route. Therefore, our protocol is obviously preferable to the practical applications.

i.e. the processes and cost are very competitive to push this technique to be applied in the field of fast, flexible and portable detection. Furthermore, the operation of this detection is very fast, and can be completed within 10-20 min, which is also another factor to push this technique to be applied in market. The current strategy is believed to be very applicable more general sensing devices in such as fluorescent, Raman and infrared spectroscopies, for a various applications in biomedicine, surface and chemical analysis, food safety and environment pollutants

I hope, with above explanations, I have responded the question-1 from reviewer-1.

2. The experimental part is not clear about the way through which the laser beam is placed in the gap between particles, and this should be clarified in the manuscript.

Response: we appreciate the suggestion on the details of detection process, thus, in the revised version, we added these information.

In the current protocol, the buoyant particulates have a particle size around 30-50 μm . Thus, during evaporation of the droplet, we used a portable optical microscopy to assist the observation of the drying and particle landing position on slippery surface. We measured the SERS signal at the particle-particle interface with a Raman microscopy system. Actually, from supporting information of the initial submission, we have shown that the detection signal repeatability can also be improved by optimizing the size of laser spot (using 10X object) and laser power (Supplementary Fig. 19).

Extended Data Fig. 19 | The Raman spectra with current buoyant particulate-based SERS protocol under various laser spot sizes and powers.

3. The section entitled “Droplet drying processes in buoyant particulate system” is too long and very much unclear, and also a long explanation is devoted in the supplementary materials.

Response: we thank the suggestion, and in the revised version, we have shortened the input of this part in the main text, and tried to improve the presentation. In the supplementary materials, in order to clearly understand the theory part of this section for reader, we kept the explanation in the supporting information as the second attachment.

4. The probe molecules employed in the so-called “Plasmonic sensing properties” were mainly colorants, such as CV or R6G. These molecules give rise to very high sensitivity (10-18 M). Additionally, other more interesting molecules such as POPs rendered also a high sensitivity (less than ppb). This sensitivity clearly is in the range of the single molecule detection. It is difficult to believe that such a high sensitivity could be got only by a simple rearrangement of the nanoparticles from coffee ring to large particle gaps where a large plasmonic effect is not expected at all because of the macroscopic character of this nanoparticle redistribution.

Response: thanks for the comments. As the explanation of question-1, the highly sensitive detection in this work can be attributed to two factors. One is the enrichment and localization effect of the detection molecules into the particle-particle interface. If using a substrate like Si surface, the droplet, e.g. 0.5 ml solvent may spread and dry on a 4-6 mm area. While, using our current strategy, we may concentrate the molecules initially in a 0.5 ml solvent and localize to the few tens micrometer area as shown in below images.

Extended Data Fig. 18 | Geometric model for the evaluation of enrichment factor.

The other factor is the plasmonic effect to enhance the SERS signal of detection molecule. Therefore, in this study, we evaluated the common used molecules such as CV or R6G, so that we can know what sensitivity we can get for the normal used molecules. And to explore the

application of current protocol in environmental contamination detection, we evaluate the capability of inspecting in persistent organic pollutants (POPs), the results display also a quite high sensitivity.

5. Finally, another discussible issue of this paper is the fact that no bands coming from the chemical reactive employed in the fabrication of the nanoaprticles and the linking system did not appear. For instance, the citrate bands, the AMPTS, etc. did not appear while others like Bisphenol A appears at a very strong intensity.

Response: We appreciate the good question from reviewer. To reply this question, we have supplemented additional data as shown below,

In fact, to obtain the silica-coated Au shell particles, a seed mediated growth method was used. The typical steps are include:

The chemical modification of hollow SiO₂ microspheres using APTMS. Here, the excess APTMS was removed by cleaning process. Then the adsorption of Au seeds on hollow SiO₂ microspheres and following removal of excess Au nanoseeds by cleaning process. Then the growth of Au nanoparticles. Actually, after growth of Au nanoparticles, the initial chemical decoration APTMS molecules and the citrate molecule from Au seed synthesis have been coated by Au nanoparticle layer. Thus, from SERS detection, we did not observed the peaks of these molecules in the interval steps of the syntheses.

From literatures, for example, the Au or Ag nanoparticles prepared by citrate reduction, similarly to the current experimental process, the interval molecules are always absent or showing a low coverage in the final product, and from sensing detection, they show a very weak even no signal. As below literatures.

1. Shikuan Yang, et al. Ultrasensitive surface-enhanced Raman scattering detection in common fluids. PNAS, 2016, 113, 268-273.
2. Li Qiang Lu, et al. Hydrophobic Teflon films as concentrators for single-molecule SERS detection. Journal of Materials Chemistry, 2012, 22, 20986.
3. Tian Li, et al. Liquid-state quantitative SERS analyzer on self-ordered metal liquid-like plasmonic arrays. Nature Communications, 2018, 9 (DOI: 10.1038/s41467-018-05920-z).
4. Zhen Liu, Zhongbo Yang, Bo Peng, Cuong Cao, Chao Zhang, Hongjun You, Qihua Xiong*, Zhiyuan Li*, Jixiang Fang*. Highly Sensitive, Uniform and Reproducible Surface-Enhanced Raman

Spectroscopy from Hollow Au-Ag Alloy Nanourchins, *Advanced Materials*, 2014, 26(15), 2431-2439

5. Dongjie Zhang, Hongjun You, Lei Yuan, Rui Hao, Tao Li, Jixiang Fang*. A novel hydrophobic slippery surface-based SERS platform for ultra-sensitive detection in food safety. *Analytical Chemistry*, 2019, 91 (7), 4687–4695

6. Kun Zhang, et al. Synthesis of micro-sized shell-isolated 3D plasmonic superstructures for in situ singleparticle SERS monitoring. *Nanoscale*, 2016, 8, 7871–7875.

7. Jin-Kyoung Yang, et al. Single Step and Rapid Growth of Silver Nanoshells as SERS-Active Nanostructures for Label-Free Detection of Pesticides. *ACS Appl. Mater. Interfaces*, 2014, 12541.

Particularly in the paper “ 8. Mengxi Xu, Bin Ren, et al. In Situ Imaging of Live-Cell Extracellular pH during Cell Apoptosis with Surface-Enhanced Raman Spectroscopy. *Anal. Chem.* 2018, 90, 13922–13928.” the citrate bands, the AMPTS, and Hydroxylammonium chloride are simultaneously used as synthetic chemical, however, in the final product, there are not peaks appear from these intermediated chemical in the detection of SERS.

Reviewers' comments:

Reviewer #2 (Remarks to the Author):

In this manuscript, the authors report a strategy to enrich and trap an analyte at a "hot-spot" for increased SERS and fluorescence sensitivity. For this they utilize, dynamics in a drying droplet. Although observed phenomenon and its utility for the goal is interesting, the conclusions are stretched too far. First of all, such sensitivity can be achieved on silver nanoparticles (AgNPs). The second issue is that the test analyte, crystal violet (CV). It is as fact that this molecule has tremendous SERS activity and the results with this molecule without testing another analytes, is quite misleading.

As the authors stated themselves, this is one of the other proof of concept reports that may not go too far.

Therefore, I cannot recommend the acceptance of this manuscript in this journal.

Response: We are grateful to the helpful comments, and the acknowledgement on "the observed phenomenon and its utility for the goal is interesting". Owing to possible reason that manuscript is not well-written and makes the readability somewhat difficult, therefore, it seems we make reviewer-2 confused in understanding the significance and the novelty of this work.

Here, I would like to explain firstly, it seems not necessary to compare current protocol and materials with silver nanoparticles (Ag NPs), since this work is mainly in the methodology of molecule enrichment and localization, not to optimize which kinds of materials or morphologies to improve sensitivity of the sensor. Actually, if we optimize the morphologies of gold, or if we use Ag NPs in current buoyant particulate strategy, a better property could be further achieved. In addition, Ag and Au have their own superiority, compared with Ag, Au nanostructures have a better stability and reliability, these could be very important for the future applications of this technique. In addition, in the field of biology and medicine, Au has a better biocompatibility.

Regarding to the second question, concerning the test analytes, in this work, totally we detected seven molecules, including crystal violet (CV), R6G, Rhodamine B (RB), Malachite green (MG), and POPs molecules including bisphenol A, 2, 4-dichlorophenol, and naphthalene. We detected CV molecule, because this molecule was always used as a standard in many papers to compare with other work. The similar high SERS active molecules have been used as probe in many high level journals such as NATURE MATERIALS DOI: 10.1038/NMAT4957; NATURE COMMUNICATIONS | 6:7800 | DOI: 10.1038/ncomms8800; NATURE COMMUNICATIONS | 8:14903 | DOI: 10.1038/ncomms14903; ACS Nano 2011, 5(6), 4407, and so on. To evaluate the current sensing protocol, and have also measured other molecules including the environmental contamination molecules, such as the persistent organic pollutants (POPs), the results display also a quite high sensitivity compared with literatures.

In this work, we put forward a new sensing strategy, we called as "Buoyant particulate strategy" for dimer or single particle-based SERS and fluorescent detections. We presented a couple of experimental and theoretical data to proof the enrichment and lifting capability of detection molecules. However, in the main text, we have also figured out the perspective of this work. For example, we mentioned that "the current buoyant particulate strategy is believed to be

applicable more wide sensing devices in such as fluorescent, Raman and infrared spectroscopes for a cost-effective, simple, fast, flexible and portable detection, and to open new possibilities for a wide range of applications.

Here, we present again the significance and novelty of this work, and its application perspective. With this buoyant particulate strategy, we may obtain below several significances,

(1) Our study represents a significant new concept on how to avoid the coffee ring phenomenon for the application of nanoparticles.

Coffee-ring effect is a very common phenomenon. In 1997 (nature, 1997, 389, 828), researcher reported the nature of coffee ring, which is the capillary flow outward from the centre of the drop carries dispersed particulates to the edge as evaporation proceeds. In 2011 (nature, 2011, 476, 308), researcher found the “suppression of the coffee-ring effect by shape-dependent capillary interactions”.

Why we are talking about the coffee-ring effect here?

Since for the applications of NPs, such as in any kinds of nanosensing used for molecule detection, the uniformity of a substrate is so so important. See above example (RSC Adv, 2013, 3, 17829). From A, for 1, 2, and 3 points, point-3 shows better signal, while, for B, normally point-1 should be better than point-2, but the results are quite different. The signal repeatability is a main problem for many kinds of applications of NPs, which is one of the key factor that block their practical applications.

Therefore, in the last decades, you may find **so many researches were carried out to fabricate array structures** by means of nanofabrication routes such as lithography or template. On the other hand, the **self-assembly of NPs** is also to solve the structural uniformity issue when using NPs as building units.

In plasmonic sensing researches, including SERS detection, the formation of coffee-ring may result in completely uncontrolled distribution of both colloidal nanoparticles and target molecules, thus a worse signal uniformity and sensitivity.

In this work, large-sized, light-weight floatable particles are designed. During the drying process, the buoyant particles may float on the top of the solvent, thus significantly reduce the chance of being pinned on slippery surface. The buoyant particulates, like a micro-sized “boat”, finally land on the substrate. **As a result, the coffee-ring phenomenon has been fully avoided.**

(2). The enrichment or localization of detected/target molecules is very important topics for many important sensing applications.

In current work, we used large-sized floating particles combining with slippery surface may enhance the spatial enrichment and localization capability of the analyte into plasmonic sensitive sites via the aggregation and lifting effect, as shown in below fluorescent images.

As a result, we achieved an excellent enrichment capability with an **enrichment factor up to an order of $\sim 10^4$** . That is to say, we concentrate the molecules initially in a 0.5 ml solution and localize to the few tens micrometer area as shown in below images.

(3). The property of current sensing strategy is very sensitive, i.e., for enhanced Raman and fluorescence spectroscopes.

Due to the excellent enrichment capability, the limit of detection for most of molecules may down to **fM level with 100% probability** to collect observable signal. Even for the **typically weakly adsorbed molecules, a limit of detection of ~ 0.1 ppb level has been easily obtained**. These results are quite few if compared with other techniques in the literatures, even at least three or four orders of magnitude higher than that of marketing product.

(4). We used this significant concept and methodology of this work with a very facile, simple, and cost-saving route. Therefore, our protocol is obviously preferable to the practical applications.

i.e. the processes and cost are very competitive to push this technique to be applied in the field of fast, flexible and portable detection. Furthermore, the operation of this detection is very fast, and can be completed within 10-20 min, which is also another factor to push this technique to be applied in market. The current strategy is believed to be very applicable more general sensing devices in such as fluorescent, Raman and infrared spectroscopies, for a various applications in biomedicine, surface and chemical analysis, food safety and environment pollutants

To our knowledge, the current results are quite valuable and represent outstanding advance and open new possibilities for a wide range of applications not only in plasmonic enhanced spectroscopy, but also in biological sensors, printing, photonic crystals, complex assembly and

other devices. Therefore, we believe this work will attract a very wide attention from related science and technology community.

Reviewers' comments:

Reviewer #3 (Remarks to the Author):

The work by Zhang et al reports a plasmonic sensor, which harnesses the electromagnetic hotspots generated through buoyant plasmonic particulate strategy. This particle assembly strategy avoids the coffee-ring effect and is able to guide the target molecules into the hotspot regions to take fully advantage of the enhanced electromagnetic field to achieve highly sensitive detection of various molecules. However, the method is not practical for real-world application owing to the difficulty in accurately controlling the volume and to place the particles inside the solution. The work described in the manuscript does not significantly advance the state-of-the-art. It might be better suited for a more specialized journal (e.g., Light: Science and Applications) There are also several suggestions listed as follows:

Response: we really appreciate reviewer-3 suggested so many good comments and advices. With these good suggestions, we have supplemented several experiments in the revised version in Extended Data Fig. 15 and in Fig. 4a and b.

We have add new results in the revised version where we realized the basic control of the accurate volume containing the floating particles, thus we may better sort the floating particles from suspension. Thus we can simply operate the detection processes by means of two steps: First, sort floating particles by pipette with an accurate volume (e.g. 5 ul) from suspension and place the droplet on the slippery surface. Since the solvent volume now can be accurately control by the pipette, Thus, with above operation, we have obtained sorting of one or two particles from suspension with a ratio of 70-80%. Actually, by simply operate using pipette, the accurate control of liquid volume has been improved compared with the superhydrophobic-superhydrophilic array in the initial version. Second, we add solvent containing detection analyte into the first droplet. Following with the assistance of portable optical microscopy, we can confirm the particle amount in the big droplet. We can complete the SERS detection within ~ 5 min for ethanol solvent and ~15 min for water solvent.

Fig. 4 | The superhydrophobic slippery surface protocol to sort single and dimer buoyant particulate. a, the droplet with a certain volume on superhydrophobic slippery surface. **b,** the optical images of sorting single and dimer buoyant particulate on the top of droplet.

Regarding the accurately controlling the volume and placing particles inside the solution or onto the slippery surface, in the real-world application, it could be worked as below method: we put the sorted floating particles with a small volume of suspension into a container (as shown in below picture). Like the SERS chip in the market, one chip for one SERS measurement. When we detect a sample, we only need place this small volume of suspension containing the selected floating

particles onto the slippery surface, and adding the solvent containing target molecule into this droplet, and evaporate them, then measure with Raman spectroscopy.

Extended Data Fig. 15 | Manipulation procedure of single buoyant particle by hydrophobic slippery substrate. Route I: after sorting of the particle by hydrophobic surface, then confirming the amount by microscope, and directly adding the solvent containing analyte. Route II: after sorting and confirming the amount of the floating particles. The particles and few solvent are stored in a container for the usage when detection.

Here, actually, we focus our work is to report a highly sensitive sensing strategy, and the main data are used to present the enrichment and lifting effect of the solvent and target molecule, hence the benefits from these effects for SERS and fluorescence spectroscopy.

We believe, when we consider and explore the real-world application using this technique, the sorting of floating particles could be not a big deal, if using the advanced sorting equipment shown above or via microfluid technique (as shown in above picture).

http://www.bio-goods.com/products_bio/fabu/SupplyItem1541575425328.html

http://www.bioon.com.cn/product/show_product.asp?id=369474

<https://www.namocell.com/single-cell-dispensers/>

Now, let me to explain the current work represents several significantly advance in below serval points:

(1) Our study represents a significant new concept on how to avoid the coffee ring phenomenon for the application of nanoparticles.

Coffee-ring effect is a very common phenomenon. In 1997 (nature, 1997, 389, 828), researcher reported the nature of coffee ring, which is the capillary flow outward from the centre of the drop carries dispersed particulates to the edge as evaporation proceeds. In 2011 (nature, 2011, 476, 308), researcher found the “suppression of the coffee-ring effect by shape-dependent capillary interactions”.

Why we are talking about the coffee-ring effect here?

Since for the applications of NPs, such as in any kinds of nanosensing used for molecule detection, the uniformity of a substrate is so so important. See above example (RSC Adv, 2013, 3, 17829). From A, for 1, 2, and 3 points, point-3 shows better signal, while, for B, normally point-1 should be better than point-2, but the results are quite different. The signal repeatability is a main problem for many kinds of applications of NPs, which is one of the key factor that block their practical applications.

Therefore, in the last decades, you may find **so many researches were carried out to fabricate array structures** by means of nanofabrication routes such as lithography or template. On the other hand, the **self-assembly of NPs** is also to solve the structural uniformity issue when using NPs as building units.

In plasmonic sensing researches, including SERS detection, the formation of coffee-ring may result in completely uncontrolled distribution of both colloidal nanoparticles and target molecules, thus a worse signal uniformity and sensitivity.

In this work, large-sized, light-weight floatable particles are designed. During the drying process, the buoyant particles may float on the top of the solvent, thus significantly reduce the chance of being pinned on slippery surface. The buoyant particulates, like a micro-sized “boat”, finally land on the substrate. **As a result, the coffee-ring phenomenon has been fully avoided.**

(2). The enrichment or localization of detected/target molecules is very important topics for many important sensing applications.

In current work, we used large-sized floating particles combining with slippery surface may enhance the spatial enrichment and localization capability of the analyte into plasmonic sensitive sites via the aggregation and lifting effect, as shown in below fluorescent images.

As a result, we achieved an excellent enrichment capability with an **enrichment factor up to an order of $\sim 10^3$ - 10^4** . That is to say, we concentrate the molecules initially in a 0.5 ml solution and localize to the few tens micrometer area as shown in below images.

(3). The property of current sensing strategy is very sensitive, i.e., for enhanced Raman and fluorescence spectroscopes.

Due to the excellent enrichment capability, the limit of detection for most of molecules may down to **fM level with 100% probability** to collect observable signal. Even for the **typically weakly adsorbed molecules, a limit of detection of ~ 0.1 ppb level has been easily obtained**. These results are quite few if compared with other techniques in the literatures, even at least three or four orders of magnitude higher than that of marketing product.

(4). We used this significant concept and methodology of this work with a very facile, simple, and cost-saving route. Therefore, our protocol is obviously preferable to the practical applications.

i.e. the processes and cost are very competitive to push this technique to be applied in the field of fast, flexible and portable detection. Furthermore, the operation of this detection is very fast, and can be completed within 10-20 min, which is also another factor to push this technique to be applied in market. The current strategy is believed to be very applicable more general sensing devices in such as fluorescent, Raman and infrared spectroscopies, for a various applications in biomedicine, surface and chemical analysis, food safety and environment pollutants

To our knowledge, the current results are quite valuable and represent outstanding advance and open new possibilities for a wide range of applications not only in plasmonic enhanced spectroscopy, but also in biological sensors, printing, photonic crystals, complex assembly and other devices. Therefore, we believe this work will attract a very wide attention from related science and technology community.

Particularly, referee asked the new evident in question-2, we provide more strong evidence to support the enriching and lifting effect of the solvent from substrate in the revised version. See below,

1. The particulate system is not well characterized. What is the structure of the particles before and after Au deposition? SEM images and TEM images are required to show the buoyant plasmonic nanostructures and to show the size of the AuNPs on the buoyant particle. What are the optical properties of such particles? For example, localized surface plasmon resonance wavelength of the particles critically determines their SERS activity for a given excitation wavelength. So the authors should better characterize the materials. How does the size of the AuNP effect the final enhancement efficiency?

Response: we appreciate reviewer's good suggestions. In the revised version, we reedited the characterization section, and added new data to clarify these questions in the supporting information.

Extended Data Fig. 1 | The schematic procedure for the preparation process of buoyant SiO₂-Au core/shell particulate.

Extended Data Fig. 2 | (a) The SEM image of hollow SiO₂ particles. (b) The TEM image, the size distribution and the UV spectrum of Au seeds. (c) The SEM images of colloidal Au seeds decorated on hollow SiO₂ surface.

Briefly, we reedited Extended Data Fig. 1, Fig. 2, and Fig. 3 as follows, and added the TEM image of Au seeds, and the size distribution chart. Additionally, we synthesized new size of Au NPs coated on silica surface. Then we characterized the scattering spectra and Raman spectra for the buoyant plasmonic particulate with different sizes of Au NPs. From the SERS measurements, we can see that with a larger particle size of Au NPs, and 785 nm laser irradiation, the SERS performance can

be improved. Concerning the size of the Au NPs effect the final enhancement efficiency, in the literatures, e.g. J. Raman Spectrosc. 2008; 39: 1679–1687, and recent paper from my group Journal of Materials Chemistry C, 2019, 7, 15259 – 15268, both demonstrated this important point.

Since, in current manuscript, we mainly in the methodology of molecule enrichment and localization. Actually, we have more opportunity to optimize the morphologies, size, structure of gold NPs, and achieve an improved sensing performance.

Extended Data Fig. 3 | The SEM characterizations of the buoyant SiO₂-Au NPs particulate, where Au NPs grew on SiO₂ surface. The size of Au nanoparticles is around 57 nm.

Extended Data Fig. 20 | (a) SEM characterizations of the buoyant SiO₂-Au NPs particulate, where Au NPs grew on hollow SiO₂ surface. The size of Au nanoparticles is around 80 nm. (b) The scattering spectra for silica decorated with Au seeds, and further grew Au NPs with sizes of 57 nm and 82 nm. (c) The Raman spectra for above three particulates under 633 nm and 785 nm laser irradiation.

2. The conclusion in line 97-100 in page 5 needs more experimental support. It is not convincing to state that CV molecules do not absorb on the substrate but in the vicinity of particle-particle interface by simply showing the absence of fluorescence signal after removing the particles. In fact, this argument is one of the main advantages the author claimed in the paper. However, the non-detectable fluorescence signal on the substrate might due to the absence of plasmonic enhancement.

Response: we really appreciate this good question raised by reviewer. Yes, this effect is really one of the main advantages in the paper since this effect determine the enrichment and localization capability of the target molecule.

Therefore, in the initial version of the manuscript, we use Figure 3 and many texts in supporting information. From theoretically, presented that, with a large particle size, e.g. larger than 30 micrometer, the aggregation effect and lifting effect could be more favorable. And from experimentally, we also showed that after movement of the aggregate, we did not found the fluorescence signal of CV molecule.

However, above experimental results could be a little weak to eliminate the contribution from noble metal enhancement. Therefore, in the revised version, we don't use the SiO₂ particle core coated with Au nanoparticles, but the pure hollow SiO₂ particle. Thus, without the plasmonic enhanced effect, whether we can still observe the fluorescent signal of CV molecule?

Extended Data Fig. 7 | Optical and fluorescent images of aggregate using hollow SiO₂ particles with CV concentration of 10⁻⁸ M. The red circles show that after we moved the aggregate, the original site does not display the 'network'-shaped fluorescent image.

Extended Data Fig. 8 | Fluorescent images of aggregate after a droplet evaporation on the slippery surface containing the CV molecules with concentration of (a) 10⁻⁸ M and (b) 10⁻⁹ M. The detectable fluorescent signal can still be observed at a CV concentration of 10⁻⁹ M.

In the revised version, we added this new result. From above images, we can see that, the CV molecule, at concentration of 10⁻⁸ M, mainly enriches into the particle-particle interface. And after moving of the aggregate, we really don't found any molecule residua on the slippery surface.

Meanwhile, when we even don't use the hollow SiO₂ particles to promote the enrichment, if we evaporate a droplet containing the CV molecule on the same slippery surface, we can still see a fluorescent image, at the CV concentration of 10⁻⁸, even 10⁻⁹ M. This comparison experiments indicate that in **Extended Data Figure 7d**, after moving of the aggregate, the CV concentration at the original site could be the level of less than 10⁻⁹ M even below.

Above supplementary experiments indicate that, using buoyant SiO₂-Au core/shell particle strategy, the CV molecule at the end of evaporation, can mainly enrich and localize into the particle-particle interface region.

3. Fluorescence quenching effect needs to be considered when calculating the fluorescence enhancement factor (page 5, line 111-114) by comparing the signal between the interface and non-interface regions. Due to the direct contact between the fluorophores and the Au surface, fluorescence signal from the fluorophore might be quenched. Therefore, the measured enhancement factor might need to be modified.

Response: we thank reviewer raising this question. We made a mistake in the description of this paragraph. Actually, here, we don't like to provide a calculated enhancement factor for SERS or fluorescent spectroscopies, but we would like to compare the signal intensity in the particle-particle interface region with the non-interface region. Thus, in the revised version, in order to avoid the possible misunderstanding, we replace the fluorescent spectra by the Raman comparison.

About the fluorescence enhancement factor, the quenching effect does exist in the bare noble metal surface. We studied below literatures, on the calculation of fluorescent enhancement factor, it should be the ratio of: $EF = I_{\text{enhanced}} / I_{\text{non-enhanced}}$. While, to get without quenching, an inert shell, e.g. SiO₂, should be used. Therefore, in this manuscript, we only measured the detection limit for SERS and fluorescence.

If we would like to calculate an fluorescent enhancement factor, we can measure the intensity with and without Au NPs coated as shown in below curves.

1. Development of Gold Nanoparticle-Enhanced Fluorescent Nanocomposites. *Langmuir* 2013, 29, 1584–1591.
2. Tuning the Intensity of Metal-Enhanced Fluorescence by Engineering Silver Nanoparticle Arrays. *Small*, 2010, 6, No. 9, 1038–1043.
3. Preparation of Ag/Au bimetallic nanostructures and their application in surface-enhanced fluorescence. *Luminescence* 2015; 30: 1090–1093.

4. Other factors that affect the drying process might be interesting to know for the readers.

Response: we thank reviewer for the suggestions. Initially, we think about this manuscript is to mainly present the focusing idea, i.e., enrichment and localization effect of target molecules, and so far, considering there have had more than twenty Figures in the supplementary information. Therefore, we don't added additional presentations of other factors. However, considering some factors are very important and as reviewer suggested, we add some results of critical factors in the revised version. These include the heating temperature and teflon membrane uniformity. Other factors, such as the pore size of Teflon membrane has also included in the revised version.

Extended Data Fig. 19 | The SERS properties as a function of heating temperature during the evaporation of a droplet containing the buoyant SiO₂-Au NPs particulate. Above curves indicate that as the increase of heating temperature, the SERS performance don't change remarkably. However, by means of increasing the heating temperature, the operation and detection period can be significantly decreased for the real-time rapid analysis.

Extended Data Fig. 20 | Last evaporation period of dimer floating particles with the Teflon membranes with some defects such as scratch or contamination. It is noted that the Teflon membrane uniformity, lubricant and the solvent types could be the significant factors to obtain a highly localized molecule enriching effect.

Extended Data Fig. 21 | The influence of pore size of Teflon membrane on the aggregation of solvent. We found that with a small pore size e.g. $0.1\ \mu\text{m}$, the enrichment effect shows excellent, and with a large pore size e.g. $0.22\ \mu\text{m}$, the droplet aggregation is not very efficient, and showing a large area distribution of the molecule.

Extended Data Fig. 22 | The influence of lubricant type and amount on enrichment effect. In the experiments, we have optimized the lubricant by using the types of Perfluorinated lubricant (Dupont, GPL 105) and (Dupont, GPL 103). From the results, no significant influence has been found, and with GPL 103, the slippery capacity can be increased. For the amount of the lubricant, if less than a critical value, the slippery ability decreases, the enrichment effect would be worse.

5. Error bars are missing in several important plots, such as Figure 4d. The experiment needs to be repeated at least three times to show the reproducibility and determine the coefficient of variance. In other words, what is the sample-to-sample variation in this sensing system?

Response: we thank reviewer for the suggestions. We have repeated the experiments with more than five points. The data have been added in the revised version.

6. Scale bars are missing in Figure 2 (a) (b), Figure 3 (b), Figure 4 (b)

Response: we thank reviewer for the suggestions. We have made the modification in the revised version.

7. Overall, the manuscript is not well-written and makes the readability somewhat difficult. The authors should carefully proofread and minimize grammatical errors.

Response: we thank reviewer for the suggestions. We have rewrite some parts those we marked out by red words. Moreover, we have asked the helps from an English editing company and polished the manuscript for your review.

REVIEWERS' COMMENTS:

Reviewer #1 (Remarks to the Author):

In my opinion the remarks were not addressed conveniently. The methodology applied by the authors is a mere redistribution of nanoparticles on microspheres that leads to a huge concentration of nanoparticles in the gap of these spheres. In a sample built with coffee ring effect there is also a huge concentration of nanoparticles in the ring, and thus, there is a strong plasmonic effect just there. Therefore, I cannot see any high advantage in the use of microspheres to immobilize plasmonic nanoparticles. It seems to be a too sophisticated way to obtain a similar effect. I cannot recommend publication of this article in nature communications. I suggest sending it to a more specific journal.

Reviewer #3 (Remarks to the Author):

The authors have satisfactorily addressed my comments. The manuscript can be published in the present form.

Reviewer #4 (Remarks to the Author):

This paper reports the development of a new plasmonic sensing platform consisting of buoyant plasmonic micro-size particles, hydrophilic solvent droplets including target molecules and superhydrophobic surface. I think the most important novelty in this paper is that coffee ring effects could be avoided in this system because the capillary force in the vicinity of the particle-particle interface overcomes the adhesion force between the solvent and the substrate. As a result, the buoyant particles were aggregated on the middle side of the substrate after solvent evaporation, and localized analytes in plasmonic sensitive sites.

I think novelty for the design of hot spot generation is good, and experimental data strongly support authors' claim. Three reviewers already raised many critical issues, and authors carefully revised the manuscript according to their comments, and consequently, quality of the revised manuscript has been greatly improved. Therefore, I recommend this revised manuscript for publication in Nature Communications in its present form.

REVIEWERS' COMMENTS:

Reviewer #1 (Remarks to the Author):

In my opinion the remarks were not addressed conveniently. The methodology applied by the authors is a mere redistribution of nanoparticles on microspheres that leads to a huge concentration of nanoparticles in the gap of these spheres. In a sample built with coffee ring effect there is also a huge concentration of nanoparticles in the ring, and thus, there is a strong plasmonic effect just there. Therefore, I cannot see any high advantage in the use of microspheres to immobilize plasmonic nanoparticles. It seems to be a too sophisticated way to obtain a similar effect. I cannot recommend publication of this article in nature communications. I suggest sending it to a more specific journal.

Response: As I have explained in the last revised version and the response letter (as attached below). Actually, reviewer-1 did not understand this work exactly. We are not design the methodology to redistribute nanoparticles on microspheres as raised again here by the reviewer, but we are exploit a strategy with buoyant particulate method to enrich and localize the target molecules into a small space, i.e. hot spot sites.

This idea has been completely understood and acknowledged by other reviewer, even previous reviewer-3, who, at the first review, raised many serious concerns, and these questions are certainly very helpful to improve and revise the manuscript. Then after review the revised version, reviewer-3 has fully agreed and suggested to accept the manuscript.

Again, the reviewer-4 has also fully reviewed the revised version and response letters, then suggested to accept the work.

Attachment: last response in the revised version. [1. It is mentioned that “The most superiority of current strategy is the aggregation effect of the buoyant-particulate on slippery Surface. As a result, the solvent and containing analyte can also be enriched into the vicinity of buoyant-particulate aggregate”. This implies that the presence of the large silica particles induces a redistribution of the nanoparticles, but it is doubtful that this improved method could further enhance the Raman or fluorescence intensity by a mere plasmonic effect.

Response: we appreciate reviewer-1 for the acknowledgement of the work with the input of “results are interesting and encouraging for the development of highly sensitive optical sensors”. Owing to that manuscript is not well-written and makes the readability somewhat difficult, therefore, it seems we make reviewer-1 confused. In fact, the main idea is not “large silica particles induces a redistribution of the nanoparticles”, but the enrichment and localization effect of the detection molecules. Actually, in the supporting information in the initial version, we have showed the structure of the buoyant particulate as follows. We can see that the Au nanoparticles initially coated on the silica surface by the Au seed decoration and growth processes. Certainly, in the revised version, we reedited the Extended Data Fig. 1, 2, and 3, to present clearly.

Extended Data Fig. 1 | The schematic for the preparation process of buoyant SiO₂-Au core/shell particulate, via Au seeds decoration and growth of Au nanoparticles on silica core.

Reviewer #3 (Remarks to the Author):

The authors have satisfactorily addressed my comments. The manuscript can be published in the present form.

Response: thanks so much for your support to accept the work.

Reviewer #4 (Remarks to the Author):

This paper reports the development of a new plasmonic sensing platform consisting of buoyant plasmonic micro-size particles, hydrophilic solvent droplets including target molecules and superhydrophobic surface. I think the most important novelty in this paper is that coffee ring effects could be avoided in this system because the capillary force in the vicinity of the particle-particle interface overcomes the adhesion force between the solvent and the substrate. As a result, the buoyant particles were aggregated on the middle side of the substrate after solvent evaporation, and localized analytes in plasmonic sensitive sites.

I think novelty for the design of hot spot generation is good, and experimental data strongly support authors' claim. Three reviewers already raised many critical issues, and authors carefully revised the manuscript according to their comments, and consequently, quality of the revised manuscript has been greatly improved. Therefore, I recommend this revised manuscript for publication in Nature Communications in its present form.

Response: thank you for provide the beneficial comments on the revised version.